# Empowering Voice through the Creation of a Safe Space: An Experience of Aboriginal Women in Regional Queensland

**DOI:** 10.3390/ijerph17051476

**Published:** 2020-02-25

**Authors:** Christina M Bernardes, Patricia C Valery, Brian Arley, Gregory Pratt, Linda Medlin, Judith A Meiklejohn

**Affiliations:** 1Cancer & Chronic Disease Research Group, QIMR Berghofer Medical Research Institute, Herston, Queensland 4006, Australia; patricia.valery@qimrberghofer.edu.au; 2Aboriginal & Torres Strait Islander Health Research Unit–Clinical Yarning Study, QIMR Berghofer Medical Research Institute, Herston, Queensland 4006, Australia; gregory.pratt@qimrberghofer.edu.au (G.P.); linda.medlin2@bigpond.com (L.M.); 3Wellbeing and Preventable Chronic Disease Division, Menzies School of Health Research, Brisbane, Queensland 4000, Australia; brian.arley@menzies.edu.au; 4Orange Sky Australia, Albion, Queensland 4010, Australia; judith@orangesky.org.au

**Keywords:** indigenous, voices, story, empowerment, Queensland, regional, wellbeing

## Abstract

Background: Sharing stories creates a space for respectful conversation and contributes to both knowledge and a sense of fellowship. This paper reflects on the experience of the research team in supporting a group of Aboriginal women to create safe spaces and to share their stories of healing, social and emotional wellbeing. Methods: Secondary data of a study exploring community perceptions about cancer were analysed using the holistic model of Indigenous Wellbeing developed by the Rumbalara Aboriginal Cooperative (2008). Qualitative data were collected during yarning sessions with a group of Aboriginal women while creating a quilt. Results: Four elements of the Indigenous Wellbeing model were identified: connectedness, sense of control, threats and relationship with the mainstream. The yarning sessions promoted dialogue, identified needs and supported the voices of the participants. A duality of forces (positive and negative) influencing community wellbeing was identified. Conclusion: This project highlights the need for spaces that support people and the community to express concerns, identify needs, propose solutions and begin a dialogue that encourages empowerment. Community-driven conversation and the identification of safe and empowering spaces can serve to empower social and emotional wellbeing.

## 1. Introduction

Health inequalities persist for the Aboriginal and/or Torres Strait Islander peoples of Australia (respectfully referred to herein as Indigenous Australians), despite Commonwealth and State imperatives such as “Close the Gap”, 2006 [1]. Indigenous Australians comprise a diversity of distinct cultural groups, each with its own unique and varied sets of lore and cultural protocols [2]. However, with colonisation, traditional Indigenous Australian cultural systems, values and protocols have been oppressed and stories silenced in favour of a dominant western view and knowledge system [3].

The National Aboriginal Health Strategy (NAHS) defines health and wellbeing as “not just the physical wellbeing of an individual but the social, emotional and cultural wellbeing of the whole Community in which each individual is able to achieve their full potential as a human being, thereby bringing about the total wellbeing of their Community. It is a whole-of-life view and includes the cyclical concept of life-death-life” [4]. Social and emotional wellbeing is a multifaceted concept that recognises the importance of connection to land, culture, spirituality, ancestry, family and community. Changes to social and emotional wellbeing may derive from grief and loss, trauma and abuse, domestic violence, removal from family, substance misuse, family breakdown, racism, discrimination and social disadvantage [5]. There is growing recognition of the importance and capability for Indigenous perspectives to inform better practice [3] and to assure social and emotional wellbeing [5,6].

Indigenous knowledge systems have underlying epistemologies that are distinct from those of academic health sciences [7,8]. Stories themselves have the power to heal, to serve as agents for catharsis [9]. It is recognised that yarning is a conversational process that involves the telling and sharing of stories, experiences and information. Yarning is culturally ascribed and cooperative, with conversations adhering to language protocol that supports reviewing knowledge to evolve the existing and account for the new [10].

In this article, we reflect on our experience of supporting a group of Aboriginal women from a community in regional Queensland to create a safe space to share their stories with one another and, with us as privileged bystanders, to support exploration of cancer perceptions in their community and to determine how this helped to identify unmet needs and facilitate empowerment.

## 2. Materials and Methods 

Secondary data analysis was conducted on data collected to explore cancer perceptions in an Aboriginal community in regional Queensland, described elsewhere [11]. Briefly, this project was informed by community-based participatory processes [12] following an initial request for this project by a community cultural adviser. The research team organised community engagement forums to ascertain the need for a project to explore community perceptions of cancer. Further group discussions helped to explore community perceptions of cancer and identify resources that could be developed to raise awareness of cancer in the community. While analysing the data about cancer perceptions, we identified rich information on how creating safe spaces to share stories can support people in the community to voice their needs, promote dialogue and facilitate empowerment.

### 2.1. Study Participants and Recruitment

This project was conducted in a small community in Queensland, Australia, located approximately 175 km from a large regional city [13]. At the time of data collection, the community had a population of just under 1000 people, most of whom identify as Aboriginal and/or Torres Strait Islander [13]. Representatives of the community and an international non-profit organisation, as well as the cultural adviser who instigated the project, assisted in promoting the project among their respective networks. Recruitment strategies included the placement of flyers in visible areas of the community and interviews with the local radio station. Participation was open to consenting community members aged over 18 years. A core group of about 11 women aged between 22 and 75 years old participated in the creation of a quilt. The participants were community members, cancer survivors and community workers who would meet in a community women’s shelter.

### 2.2. The Process

Participatory, qualitative and visual research methods were employed to gather data. Qualitative data to explore and define cancer in the community were gathered from yarning groups, including men and women at two open community forums held at the beginning of the project, and subsequent yarning groups with women while a resource for community was being created, specifically a community quilt. Written informed consent was obtained from eligible participants prior to audio taping the yarning groups. Data were also included from publicly available interviews about the project broadcast on radio and television.

The creation of the community quilt required the participants to come together to decide on the material, messages and how to collate these messages. While weaving the quilt, the research team aimed to primarily explore community perceptions of cancer; however, the yarning sessions included broader dialogue about health, wellbeing and community. While small sections of the quilt were worked upon individually, the majority of the quilt was created and completed over six separate sessions held with the research team, and data were captured throughout these sessions.

Figure 1 illustrates the process and outcomes observed by the participants and research team.

### 2.3. Framework and Data analysis

The framework used to analyse the data was the holistic model of Indigenous Wellbeing developed by the Rumbalara Aboriginal Cooperative (2008) [14,15]. In this model (Figure 2), the authors emphasised the deep connection Indigenous people hold to their ancestral land (country) and its association with their wellbeing. The framework is not a measurement tool but rather the application of a holistic model to better understand the driving forces of, and implications for, Indigenous peoples’ wellbeing. The framework includes the representation of the multiple forces (positive and negative) that impact the wellbeing of people and community.

This holistic model includes five key elements, namely connectedness, threats, sense of control, history and relationship with the mainstream.

In the model, country includes everything from land, air, water and stories of “dreaming” and is dynamic and multi-layered, with norms and beliefs of existence connecting Aboriginal people from the time of creation. The connectedness element is subdivided: in role models, community, land and space. The threat element is composed of feelings of boredom and misuse of alcohol and drugs, multiple pressures and grief. The element of sense of control is composed of healthy thinking, personal beliefs and success and acknowledgement. The history element includes ongoing trauma and strength and triumph. The fifth element of relationship with the mainstream is composed of reconciliation and reciprocity, marginalisation and racial stereotypes.

Data were transcribed verbatim and analysed thematically [16]. Three research team members (CB, JM and LM) initiated the analysis process by coding a sample of the transcripts independently. Next, initial codes were compared for inconsistencies and intercoder reliability. Disagreements were further discussed and refined until an agreement was reached for final coding. This was then applied to the full set of transcripts. A final stage involved the mapping of the sub-themes under the five key elements (connectedness, threats, sense of control, history and relationship with the mainstream) of the model.

### 2.4. Ethics Approval

This research was approved by the Human Research Ethics Committee of QIMR Berghofer Medical Research Institute (HREC―P2237).

## 3. Results

The results presented here were collected during the various group yarning sessions while making a quilt in a community women’s shelter. In this space, the research team observed the impact on the wellbeing of the participants.

The study was developed by the community in response to an identified concern; the participants defined the aim, the process and outputs for the project. Conversations at the first meeting confirmed the guidelines for interaction and discussion. The location used by the participants was collectively identified as suitable for further meetings. This contributed to ensuring a safe space for people to meet and support one another as well as achieving the objectives of the project. The research team was transparent in their communications about the project, its purpose and responsibilities, confirming with the participants that they understood and agreed to being involved. Time was taken at the beginning of the creation of the quilt to build rapport and familiarity between the research team and the participants. Members consistently attended group meetings, reflecting that they felt the space was indeed “safe”. Interactions between members were respectful and conversations in-depth. The participants reported that they enjoyed the experience of attending the group and that they were able to make a contribution to discussions and decisions. A core group of 11 participants was present at every session and made significant contributions to discussion and action. While the purpose of the group was to discuss cancer perceptions, the participants recognised the space as a safe place to also discuss culture, identity and share their concerns for family and the community with the researchers and each other. The public information in radio and television contributed to the data gathering on cancer perceptions in the community but also demonstrated that the participants in the women’s group felt much more confident to advocate about the need for support in the event of cancer.

Of the five elements of the holistic model of Indigenous Wellbeing [14,15], four elements were recurrent in our data: connectedness, sense of control, threats and relationship with the mainstream.

### 3.1. Connectedness

The first element identified throughout the transcripts was “connectedness”, and under this element three sub-themes emerged: starting the conversation, building a sense of community and giving and receiving support. The participants reported that making the quilt created a space to start a conversation about cancer.
“Lots of our mob doesn’t know how to read and write. We learn better by doing and we hope that by doing this quilt we will start the conversation and beat the taboo about cancer.”(Community worker)
“And I think because you work together and people see you actually doing something they then can ask questions. And the two way communication starts.”(Community member)

This led them to be hopeful that the experience could promote conversation in other areas of health such as living healthy, eating healthy and participating in screening tests. Many of the participants emphasised the feeling of connection, working together and how this built a sense of community and strength.
“I think as this sort of thing, as a community I think people get together doing stuff like this that wouldn’t normally even talk to each other. And it gives like-minded people something to connect to.”(Community member)
… “It’s good to have a women’s group and a group of ladies that you know, get together and build bridges and video come out, you know build awareness programs in our community and people. Yeah. Tell them that we’re there for them you know tell them we can walk that walk.”(Community member)

The space created was also seen as a place to give and receive support for those who were affected by cancer. For some of the participants, this space helped in the healing process, where previously unspoken suffering could be shared.
“I lost a sister, a young sister to cancer of the cervix. So that was my contribution and just encouraged me to be there to support the other women.”(Community member)
“Some people sort of like to sort of I suppose suffer in silence. But if you can get them to come out it’s so much better for them if they can come out.”(Community worker)

### 3.2. Sense of Control

In the “sense of control” element, one sub-theme emerged: taking control. The participants acknowledged the importance of taking control of their health and wellbeing, taking time for themselves and obtaining information about what is happening with their health. The participants indicated that talking, asking questions and sharing experiences could improve knowledge and understanding and was also an opportunity to seek advice for health issues. This led to them feeling empowered to make decisions about their health and wellbeing.
“Because if we get sick and we find out that we’ve got that, well we can you know just sit down and talk about it.”(Cancer survivor)
“…you know some of us are after you know in our forties and we need to talk about midlife crisis you know. Menopause. Menopause you know we up in age and we need to know about that you know.”(Community member)

### 3.3. Threats

Within the “threats” element, four sub-themes emerged, namely disconnection between generations, inability to pass on knowledge, shame and fear and collective grief and loss. The participants identified a significant rupture between the younger and older generations, with a lack of interest in culture and traditional ways of teaching and learning. The participants who were Elders shared their stories of growing up in a community surrounded by culture and traditional ways of learning. When Elders were young, a fire would be lit in two camps: girls were taught by women and boys were taught by men about nature and survival. There was a great sense of responsibility to care for the community and a closeness in the way people lived.
“We had to go to school every afternoon when we finished school and even on a Saturday and Sunday and the weekends were our culture days where the women would take all the girls out. The men would take all the boy out and they teach them everything about the bush.”(Cancer survivor)

Community members were taught about culture, nature and life by doing things together. Some of the Elders expressed their frustration that this is not currently happening and reported that they missed the days when they were respected for their knowledge and could talk to younger people in the family and community about culture, traditional teachings or health issues.
“Is hard to get that thing going on again, the younger children, the teenager generation is not interested in this sort of things anymore…they are not interested in culture…”(Cancer survivor)
“Because just sit down, you can’t sit down and have a decent conversation with, I can’t sit down and have a decent conversation with my own granddaughter about those sort of things. She sits down and laugh in my face.”(Cancer survivor)
…“Is hard, you got so much knowledge, you try to drum into the young people but they don’t want to listen you…they are not interested anymore.”(Cancer survivor)

When talking about cancer, the participants mentioned that it was a topic the community did not want to talk about because of fear and shame. Some of the participants reported that shame prevented families from telling their children about the diagnosis. In turn, some families, when aware of a cancer diagnosis, chose not to talk about cancer due to fear of what the diagnosis means.
“They think of cancer as some dirty kind of disease which it is not…A dirty disease.”(Cancer survivor)
“The way the, even if somebody dies a normal death and you have a funeral here or somebody is sick up in hospital ready to die, they’ve got a lot of friends. When they’re up here in this hospital bed on their death bed they haven’t got a friend in the world. Nobody goes to see them. Even the families are too frightened to go up there and stand beside them and hold their hand. Yeah everything revolves around fear”(Cancer survivor)

With regards to the sub-theme of “collective grief and loss”, the participants indicated how major life events (e.g., birth or death) are emotionally shared across the community. For example, if someone becomes unwell, all the community members endure the burden and suffering that come with that experience.
…“They took me down to a cancer ward in Rockhampton about 10 years ago and I walked through that door. And I just, even though I had it myself, I just stood there and cried for the people I saw in those beds. I didn’t want to be in the same room with them, I wanted to be somewhere else. Because didn’t want to otherwise I wouldn’t be worrying about myself, I would have been worrying about everybody else that was in that room with me.”(Cancer survivor)

### 3.4. Relationship with the Mainstream

In this element, the sub-theme that emerged was the bridge between traditional and Western medicine. The participants talked about two aspects of their relationship with the use of traditional medicine to improve health outcomes for cancer patients and the need for Western medicine to check their health.
“That’s Gumby-Gumby tree so that’s one of our medicines here that we use. We can bath in it and it clears up your skin or you can drink it. …Well my uncle took it with him when he went down for his chemo. And he was the only one able to eat when nobody else could.”(Cancer member)
“Every one of my granddaughters now they are all young mothers now. And they I always remind them don’t forget that van is out the back of the hospital, go and get yourself checked again you know….So I tell all the granddaughters.”(Cancer survivor)

## 4. Discussion

The participants in this study reported the space created while weaving the quilt promoted reconnection with their culture and renewed their sense of belonging. This space was a break from the participants’ ordinary routine that promoted the sharing of knowledge and recollection of the traditional ways of life. Since colonisation, Indigenous Australians have lived with the consequences of losing their land, culture and way of life [17]. Interestingly, the participants identified the group as a “space”, reflecting what Dudgeon (2006) described as the reality that dispossessed people have no choice other than to make “space” in a “place” now owned and controlled by colonisers [18].

In this space, emotions and memories were triggered. The space was felt as a “recovery” space where participants could support each other, freely discuss a variety of needs that otherwise would not be talked about and promote healing. Rehabilitation and recovery programs offer a variety of activities and therapies, but for many Indigenous peoples, storytelling serves as a medium through which to learn about the different aspects of life [3,19]. Telling stories supports a person in developing and strengthening their sense of identity and belonging and in healing. Yarning is a special and powerful part of Indigenous Australian culture that connects people to one another, their spirit, history, land and spirituality [20].

Importantly, the participants identified that they learn more effectively by doing things together; the story shared by one person can enlighten others. Indigenous Australians recognise the value and meaning of their connection with family and community; an individual cannot stay well if the community around them is suffering. Sharing stories and experiences with other people serves as an opportunity for the individual to alleviate their pain—for one to heal through his or her connection with others. Individual stories of loss and resilience are also important in retaining a community narrative from past to present and into the future [21]. Having a strong social network supports a sense of social inclusion and helps reduce an individual’s feelings of isolation and vulnerability through knowing there is someone with whom to connect and by whom to feel supported [21].

The support for the wellbeing of the participants could also come from the combination of Indigenous culture and traditions and modern or Western knowledge. The participants in the study indicated that they used traditional medicine in conjunction with Western medicine. A literature review of Indigenous healing programs in the United States, Canada, Australia and New Zealand reported that combining Western methodologies and traditional healing consistently increased the efficacy of treatment [22]. The use of traditional or complementary medicine is widespread, but frequently, patients do not disclose their engagement in this practice due to concerns that it will be frowned upon by their health professional [23,24]. The use of traditional and complementary medicines by Indigenous peoples has also been noted to provide spiritual, emotional and cultural benefits [23].

A significant concern, especially among the Elders in the group, was the disconnection of younger generations with their culture. Strong cultural attachment has been found to be protective against feelings of helplessness, loss of meaning and control and to have a positive impact on educational achievement and the importance a person places on their health [25]. Elders in the group distinguished the importance of culture to build resilience and strength in the community; they described feeling anxious about not knowing how to repair this disconnection. Potentially, creating “spaces”, such as the one the women of the community created, offers younger and older an opportunity to connect with one another and to share and strengthen culture.

As previously recognised, this study demonstrated that the wellbeing of the Aboriginal women involved in this quilting group can be supported through individual, family and community connections—connections that serve to empower voice and encourage self-esteem, feelings of worth and community ownership [22].

## 5. Conclusions

Engagement with this group of Aboriginal women supported their ability to act as their own champions and to create a safe space for conversation around a taboo topic (cancer), as well as to encourage conversations and actions that they in turn were able to share with their community. The work outlined herein highlights a need for spaces that support people in sharing their voices, promote dialogue and encourage community action through empowerment. Health professionals should encourage these types of community-driven initiatives as they offer viable solutions and mechanisms to support social and emotional wellbeing. The solutions discussed in this group were simple. For example, they suggested changing the gynaecological cancer screening program from the hospital to a “friendlier” place. Health professional should be open to adjusting services to the identified needs of the community.

## Figures and Tables

**Figure 1 ijerph-17-01476-f001:**
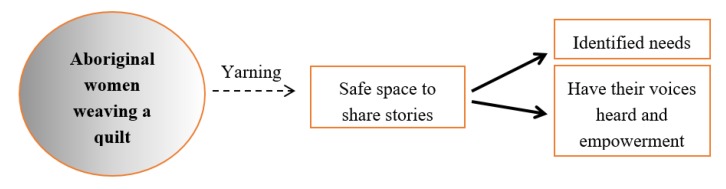
Process of data generation and observed outcomes.

**Figure 2 ijerph-17-01476-f002:**
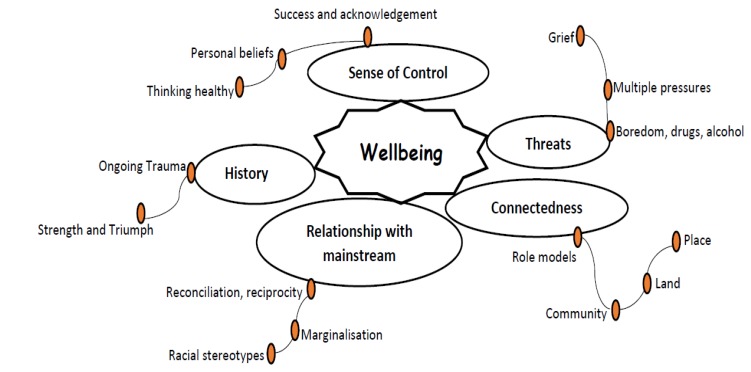
Holistic model of Indigenous Wellbeing—"Our Wellbeing” [14,15].

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
