# Peer review of "Empowering Voice through the Creation of a Safe Space: An Experience of Aboriginal Women in Regional Queensland"

_ijerph, 2020, doi:10.3390/ijerph17051476_

Round 1
Reviewer 1 Report
Empowering voice through the creation of a safe space: an experience of Aboriginal women in regional Queensland
Authors: Bernardes, Valery, Arley, Pratt, Medlin and Meiklejohn
This paper aims to describe how research-based facilitation of ‘safe spaces’ can empower Aboriginal women in a regional area to share their experiences of healing in relation to cancer, and of social and emotional wellbeing more broadly. The data used draws on secondary analysis of qualitative material collected within a broader project exploring community perceptions of cancer, especially verbatim transcripts of yarning sessions with Aboriginal women involved in a creating a quilt. A holistic model of Indigenous Wellbeing is used for analysis. The paper concludes that researcher-facilitated safe spaces can empower participants’ voices and support social and emotional wellbeing.
The Introduction draws on relevant literature regarding yarning methodology, situated within holistic conceptualisations of health that include complex understandings of Indigenous social and emotional wellbeing. However, while the Introduction seems to indicate that the focus of the paper will be on the facilitation of safe spaces and the resulting empowerment of women to share their experiences, the Results section instead appears to overly emphasise elements of the holistic model of Indigenous Wellbeing without adequately linking these elements to the proposed dynamics of safe space. This, in turn, makes it difficult to see the links between the Results presented and several assertions within the Discussion. The manuscript requires substantial reworking of the Results section in order to support the aims, discussion points and conclusions of the paper.
Several ambiguities that could be clarified include:
How is ‘safe space’ being construed? Which aspects of the forums and subsequent activity-based yarning sessions contributed to ‘safety’ – what mechanisms were put in place by researchers and community champions to create or support safety and what behaviours within the sessions indicated that this had been achieved?
o For example, in Lines 83-84, the authors state that, “Data was also included from publicly available interviews about the project broadcast on radio and television” –how do these data relate to ‘safe space’, as they appear to have been shared in public space? How do the data from the quilt-making activity differ from those collected in the other research settings?
It is unclear within Table 1 how quotes are being attributed to participants (e.g. “J-Female 1”; “R2-Evaluation”; “S4-Monday”; “R3-Evaluation”; “D-Female 3”; “S3-Monday”, etc.) – it would be helpful to add additional detail regarding the number of participants and key characteristics such as age and role in the study (e.g. cancer survivor, carer, primary healthcare professional, community champion, researcher, etc.) of the women whose data was used for this paper, and to clarify how quotes are being attributed.
o For example, from the publication regarding the overall study (Meiklejohn et al. 2019), it appears that 50 women and men participated in the original forums, with subsequent participation by 20 (gender not identified) people at the yarning groups, and 11 women participating within the women’s groups. Furthermore, several quotes that are used in Table 1 of this manuscript are presented in Meiklejohn et al. 2019 and, in that publication, attributed to ‘female health worker’ or ‘female community worker’ – therefore, clarifying whose voices are being empowered and how this relates to both ‘safe space’ and ‘wellbeing’ will be helpful.
In the Results section, the separation of Table 1, with its large list of illustrative quotes, and the headings and text describing these that are presented above the Table, make it difficult for the reader to understand how these quotes relate to either ‘safe space’ or to ‘wellbeing’. I would recommend placing the relevant quotes (and further descriptions of their relevance to each heading) under the headings within the Results section, rather than in a Table. It may also be useful to include a new section in the Results that addresses ‘safe space’, separately from ‘wellbeing’.
The quotes selected for inclusion within Table 1 are presented as illustrating four elements of the holistic model of Indigenous Wellbeing, but as a reader, they appear to be saying several different things.
o Firstly, the quotes included in the first two sections of Table 1 (i.e. Starting the conversation; Building a sense of community) and some of those in the third section (i.e. Giving and receiving support) seem to lend themselves to illustrating the creation of ‘safe space’ and the beneficial aspects of the research-based activities in supporting the sharing of stories and experiences – I would like to see the authors engage more fully with these dynamics, if this seems accurate.
o Secondly, several of the quotes included in the fifth and sixth sections of Table 1 (i.e. Rupture between generations; Can’t pass on knowledge), while currently presented as ‘threats to wellbeing’, could potentially be used to show links between the types of sharing that have been achieved through the research-based activities and older, more traditional ways of sharing knowledge and experiences between generations (e.g. within gender separate groupings) – if this was what the research team observed (see Lines 215-217). For several of the other quotes within these two sections, it is unclear how they illustrate either safe space or wellbeing – this should be made more explicit in the Results, if they are to remain included.
o Thirdly, the remaining quotes included in Table 1 may illustrate aspects of community members’ wellbeing (positive or negative) in relation to cancer that align with the holistic model of Indigenous Wellbeing, but I would like to see the authors make this more explicit within the Results text and presentation of quotes, in order to tie more closely to the points raised in the Discussion. For example, how do these quotes demonstrate the “exploration of their cancer perceptions and how this helped to identify unmet needs and facilitate empowerment” (Lines 56-57)?
The Discussion reads well; and it is likely a good summary of the research team’s observations and reflections regarding the research activities, processes and outcomes. Further description of the study participants and specific research-based activities within the Methods, as well as a more detailed exploration of the quotes presented in the Results section, will assist the reader to understand how the data presented informs the main points within the Discussion.
I really like the premise of this paper, especially as in Line 66-67 the authors observe that they, “identified rich information on how creating safe spaces to share stories can support people in the community to voice their needs, promote dialogue and facilitate empowerment”. This is interesting and potentially an excellent contribution to the literature, but needs to be better supported by the Methods and Results. I wish the authors the best.
Reviewer 2 Report
This was a well written paper on an important and interesting topic. I did look into the background to earlier work on the research design, and found this very informative.
I wondered if the authors felt they could strengthen the recommendations, or implications for health professionals.
Minor typo first sentence on abstract (word missing)
Reviewer 3 Report
Line 2 (title) – “creation of a safe space”… maybe I missed this – but it doesn’t seem to explicitly state within the paper how the safe space is created. i.e. where was the quilting actually done – in a local, familiar Aboriginal service that people regularly attended and already felt safe? What other things were done to promote safety and sharing ? Where the men and women separated for “womens business” to discuss?
Line 3 (title) – “an experience” .. should this be “experiences” as its more than one woman ?
Line 50 – “Indigenous Australians recognize yarning as…” – phrasing of this sentence needs reworking - it’s widely recognised by society that yarning is a conversational process that benefits Indigenous populations rather than just Indig pops recognising it
Lines 54 – 57 – Not clear if the women have cancer and its their perceptions on cancer or just generally their perceptions. From what it sounds like its just generally their perceptions on cancer but I would explicitly state this
Figure 1 – “have” their voices heard rather than “get” ?
Line 161 – “the” missing between topic and community
Table 1 – quotes flick between “female” and days / evaluation.. what does this mean? Why isn’t it consistent ? Also table is very long – some quotes could be easily cut in half (by adding … in the middle so the key point is pulled out)
Section 3.2 – very brief … “Participants indicated that talking and sharing could improve knowledge and was also an opportunity to seek advice for health issues.” … talking and sharing facilitated through this yarning? Or talking in general? What activities actually empowered them and allowed them to gain sense of control?
Section 3.3 – seems a bit disjointed, lots of short sentences that don’t flow on to each other and are not explained … “A fire would be lit in two camps and the traditional teaching was passed on.” – what teachings are passed on? Whats the significance? Is it different now?
“There was a great sense of responsibility for the community and closeness in the way people lived.” – "was"? So its no longer the case? Why ?
“Community members were taught about culture, nature and life by doing things together.” What things are they doing together ?
Discussion – would be helpful if the discussion had subheadings that matched the results as the themes quite different and each paragraph jumps topic / doesn’t particularly flow i.e. lines 201- 202 jumps from isolation to traditional medicine
I think overall the concept of the paper is well founded (i.e. that safe spaces are needed to help encourage the dialogue of taboo topics) – however 40% of the paper (3 pages out of 7.5) is just quotes and the explanations of why these themes are important I feel are lacking. Themes are drawn out but no information (if there is any) is provided on what was done to help i.e. shame a big issues, but is there anything being done to help empower the participants to talk about cancer more or is it just this once off yarning session ?
Reviewer 4 Report
A really interesting and evocative paper. This is an important study where participants were valued. The quotes were beautiful and made me feel at times that I was there with the women quilting too. Only minor things.
Line 70: Is it OK to identify the town? I don't think the paper loses anything by not and it then gives an extra layer of confidentiality to participants.
Lines 111-118: Beautifully explained but I wonder whether a figure illustrating these elements might help an international audience follow this better.
Line 131: Possibly just me being picky, but threats is mentioned above as the second element and sense of control as the third but in the following these are swapped around. Having threats followed by sense of control throughout would add greater cohesion in reading.
Line 176: Could participant IDs be explained in the methods section please. For example, what does Monday mean? Who was in which of the different participant groups explained?
P. 5 Giving and receiving support S5-Monday quote: Maybe a square bracket here to clarify this as an international reader might understand this as 'staying close to someone' as opposed to being closed off to people.
P. 5 Taking control R3-Evaluation quote: Place a pseudonym or something here to protect the identity just in case.
PP. 6-7: It's really interesting seeing S3-Monday have a bulk of quotes in rupture between generation and in shame and fear. Her narrative is really evocative and the stories feel very linked. T-Female-2 is also a strong voice in threats. Was there a connection between the two women who were more focused on these issues? Or were these simply the most evocative examples?

Round 2
Reviewer 1 Report
Thank you for the additional detail regarding participants, as well as the specific research activities and mechanisms that helped to create and facilitate a safe space for participants to share their experiences. The women's willingness to speak publicly on radio or television is a lovely illustration of their increasing confidence and empowerment as a result of the series of yarning sessions.
Thank you also for placing specific quotes near the descriptive content within the results section. I believe readers will now find it easier to see the links between the results and the discussion. Overall, it is now more evident how the facilitating of safe space permitted participants to share their thoughts and experiences, as well as how these relate to the identified elements within the Indigenous Wellbeing model.
I would recommend one last proof-read to catch any small typos (e.g. spacing between words or sentences resulting from tracked changes, etc.).
Reviewer 3 Report
Thank you for making suggested changes. I find the results section is much easier to follow with the addition of quotes directly into the text and the clarifications around who the participants are